# Design of an Epitope-Based Vaccine Ensemble for Animal Trypanosomiasis by Computational Methods

**DOI:** 10.3390/vaccines8010130

**Published:** 2020-03-16

**Authors:** Lucas Michel-Todó, Pascal Bigey, Pedro A Reche, María-Jesus Pinazo, Joaquim Gascón, Julio Alonso-Padilla

**Affiliations:** 1Barcelona Institute for Global Health (ISGlobal), Hospital Clínic - University of Barcelona, 08036 Barcelona, Spain; mariajesus.pinazo@isglobal.org (M.-J.P.); quim.gascon@isglobal.org (J.G.); 2Centre National de la Recherche Scientifique (CNRS), Institut National de la Santé et de la Recherche Médicale (INSERM), Université de Paris, UTCBS, F-75006 Paris, France; pascal.bigey@parisdescartes.fr; 3PSL University, ChimieParisTech, F-75005 Paris, France; 4Laboratory of Immunomedicine, Faculty of Medicine, University Complutense of Madrid, Ave Complutense S/N, 28040 Madrid, Spain; parecheg@med.ucm.es

**Keywords:** African animal trypanosomiasis, epitope-based vaccine, B-cell epitopes, CD4 T cell, CD8 T cell, pDNA, string-of-beads

## Abstract

African animal trypanosomiasis is caused by vector-transmitted parasites of the genus *Trypanosoma*. *T. congolense* and *T. brucei brucei* are predominant in Africa; *T. evansi* and *T. vivax* in America and Asia. They have in common an extracellular lifestyle and livestock tropism, which provokes huge economic losses in regions where vectors are endemic. There are licensed drugs to treat the infections, but adherence to treatment is poor and appearance of resistances common. Therefore, the availability of a prophylactic vaccine would represent a major breakthrough towards the management and control of the disease. Selection of the most appropriate antigens for its development is a bottleneck step, especially considering the limited resources allocated. Herein we propose a vaccine strategy based on multiple epitopes from multiple antigens to counteract the parasites´ biological complexity. Epitopes were identified by computer-assisted genome-wide screenings, considering sequence conservation criteria, antigens annotation and sub-cellular localization, high binding affinity to antigen presenting molecules, and lack of cross-reactivity to proteins in cattle and other breeding species. We ultimately provide 31 B-cell, 8 CD4 T-cell, and 15 CD8 T-cell epitope sequences from 30 distinct antigens for the prospective design of a genetic ensemble vaccine against the four trypanosome species responsible for African animal trypanosomiasis.

## 1. Introduction

African animal trypanosomiasis (AAT) is a debilitating disease that affects wildlife and domestic animals in vast areas of Africa, Latin America, and Asia [1]. Known as nagana or surra, it can be caused by different protozoan parasite species of the genus *Trypanosoma* (class Kinetoplastea; family Trypanosomatidae), which are transmitted by insect vectors. Pathogens and vectors co-evolved and their distribution varies geographically. *Trypanosoma congolense* and *T. brucei brucei*, the most studied of the four, are transmitted by hematophagous tsetse flies (genus *Glossina*), and along with mechanically transmitted *T. vivax* are abundant in Africa [1]. *T. evansi* is also mechanically transmitted by fly bites and along with *T. vivax* can be found in America and Asia [1]. Nevertheless, they all thrive in mammalian reservoirs free-living in their bloodstream and manage to persist there causing chronic infections. In livestock, these infections produce anemia, poor condition and reproduction, and death, which leads to huge economic losses [2]. In Africa, AAT exerts a devastating toll and it is considered the most important disease driven limitation to livestock production [3].

Infections can be treated with the drugs diminazene aceturate and isometamidium chloride, but the appearance of resistances is common [4,5]. A prophylactic vaccine used alongside vector control strategies would be a major breakthrough towards controlling the disease impact, but efforts in this direction have had limited success and there is none licensed (reviewed in [6]). The plethora of immune evasion mechanisms described in trypanosomes do not allow optimism in this respect [7], but naturally acquired immunity against trypanosomiasis has been reported in humans and animals [4,8]. In addition, some protection has been determined in AAT pre-clinical models of infection upon immunization with recombinant invariant parasite antigens like β-tubulin [9] or with a plasmid encoding an invariant surface glycoprotein (ISG) [10]. The immunization with irradiated trypanosomes or their variant surface glycoproteins (VSGs) has also been described to protect cattle against an homologous challenge (reviewed in [11]). Moreover, in a biologically relevant cow infection model, trypanotolerant N´Dama breeds have higher antibody levels in comparison to trypanosusceptible Boran cattle [11]. Thereby, it should be possible to elicit specific immune responses that protect against AAT infections, and if these immunizations were elicited by invariant conserved antigens, such responses could protect against heterologous infections, too.

In light of the high biological complexity of the parasites, and the very limited resources dedicated to the development of vaccines for AAT, a reverse vaccinology approach could be the most adequate strategy to follow. Herein we have used the genomic information available on AAT-causing *Trypanosome* spp. (downloaded from TriTrypDB release version 39) [12], and a series of tailor-made and publicly available immunoinformatic tools, to provide a list of predicted epitope sequences that could be the basis of an epitope-based vaccine. Such a vaccine could be delivered as a genetic construct at a fraction of the production and deployment costs of other vaccination strategies. The criteria employed to select the epitopes included a strict sequence conservation, sub-cellular location and annotation of their antigens of origin, high binding affinity to antigen presenting molecules, and lack of cross-reactivity to other proteins present in cows, sheep, goats, and pigs. As a result, we ultimately prioritized 31 B-cell, 8 CD4 T-cell, and 15 CD8 T-cell epitope sequences from 30 distinct antigens for the design of an ensemble pan-AAT vaccine. The rationale of choosing a plasmid DNA (pDNA) construct for its delivery is also discussed.

## 2. Materials and Methods

### 2.1. Collection of Protein Sequences from Trypanosome Spp. That Cause AAT

We downloaded the whole annotated proteomes of all *Trypanosome* spp. that have been related to AAT from TriTrypDB (release v39) [12]. We merged them to obtain a single file containing all annotated protein-coding sequences (CDS) available at the TriTrypDB resource from *T. brucei* Lister427, *T. brucei* TREU927, *T. congolense* IL3000, *T. evansi* STIB805, and *T. vivax* Y486.

### 2.2. Generation of Clusters

We used CD-HIT, running stand-alone with default settings, to reduce the redundancy of our *Trypanosome* spp. protein collection, clustering them with a shared identity >90% [13]. The resulting 28,709 clusters were filtered down by means of custom scripts to retain only those that contained at least one protein sequence from each of the 5 proteomes included in the analysis. This way we ensured that clusters considered for further analysis represented the protein diversity amongst all available sequences. Protein sequences within each of the remaining 115 clusters were aligned using MUSCLE (software version 3.8.31) [14] to obtain an equivalent number of multiple sequence alignments (MSA) with their corresponding consensus sequences. Nonetheless, instead of the consensus sequence that is built with pieces from different sequences and thus being unreal, we kept as a reference of each MSA the sequence in the alignment that was most similar to the consensus one.

### 2.3. Generation of the Invariable Proteome of Trypanosome Spp. Responsible for AAT

The Shannon entropy (*H*) measure [15] was used to assess sequence variability on every position of the MSA according to the equation (Equation (1)):
(1)$$H=-\sum_{i=0}^{M}P_{i}\log_{2}P_{i}$$
where P_i_ is the frequency of an amino acid of type i, and M is the number of totally different amino acids (20). H = 0.0 is equivalent to no variation in a given position among the studied protein sequences within the alignment, whereas higher values correspond to higher variation in that position [16]. Gaps were considered as data. Those residues at positions with an entropy value of H > 0.0 were masked in the assigned protein reference sequence, substituting the residue symbol by an asterisk symbol (*). As a result, we obtained a masked FASTA file which represented the conserved proteome of AAT trypanosomes, which we used to predict conserved epitopes within.

### 2.4. Prediction of B-Cell Epitopes

We identified the invariant regions of at least 15 residues long in the masked proteome FASTA file and created another FASTA file with the sequences of their corresponding antigens of origin. We then curated this list to retain only those fragments whose antigens of origin were expected to be surface-exposed accordingly to their annotation, as well as those whose antigens were not predicted with high reliability to be part on intracellular proteins. BepiPred2.0 was run on the whole protein sequences of the curated list with a threshold set at >0.6 [17]. Finally, we crossed the results of this prediction with the conservation results obtained from the previous procedure (see Section 2.3). As a result, only those regions of at least 15 residues long, and that were predicted as epitopes by Bepipred2.0 and conserved accordingly to our analysis, were considered as putative epitopes for subsequent prioritization.

### 2.5. Prediction of T-cell Epitopes

For the prediction of CD4 T-cell epitopes we used the IEDB recommended method, the MHC II binding predictions tool (http://tools.iedb.org/mhcii/) [18], over the invariant proteome file. Given that no bovine lymphocyte antigen (BoLA) alleles are available for this tool we ran the program against the set of human leukocyte antigens (HLAs), as described elsewhere [19]. We considered for further analysis only the 0.01% top scoring CD4 T-cell epitopes that were predicted on the *T. cruzi*-masked proteome file.

For de novo prediction of CD8 T-cell epitopes we used the IEDB recommended method, the MHC I binding predictions tool (http://tools.iedb.org/mhci/) [18]. We ran the program using the available set of BoLAs (see Appendix A) alleles over the invariant proteome file. We considered for further analysis only the 0.1% top scoring predicted epitopes.

### 2.6. Blast Searches for Epitopes Identity and Other Analysis Procedures

In order to find out the epitopes identity (i.e., % identity over the queried sequence length) to proteins in cows, we blasted them against cow microbiome protein sequences obtained from the Cow Microbiome Project [20] and against the NCBI non redundant (nr-) collection of *Bos taurus* and *Bos indicus* proteins. We used the NCBI protein blast suite (BLASTP) with the default parameters, except for the PAM30 Scoring Matrix and an expectation value (E-value) of 10,000, as the epitope sequences to blast were shorter than 100 residues in all cases. Following the same procedure, we also blasted the epitopes sequences against the NCBI nr- collection of sheep (*Ovis aries*; tax id: 9940), goat (*Capra hircus*; tax id: 9925), and pig (*Sus scrofa*; tax id: 9825) proteins. 

The prediction of patterns related with surface exposure of those antigens with predicted B-cell epitopes was made with SignalP, TargetP, and TMHMM [21]. The possible presence of glycosylphosphatidylinositol (GPI) anchor signals was predicted using the PredGPI website [22].

We used the RaptorX web portal to model the 3D structure of the vaccine ensemble peptide sequence [23]. Solvent accessibility (Acc) classifiers were obtained at RaptorX, too. PyMOL Molecular Graphics System, Version 1.8 Schrödinger, was used to visualize the ensemble 3D structure.

## 3. Results

### 3.1. Identification of the Invariant Proteome of AAT-Causing Trypanosomes

We used all the available annotated proteomes from *Trypanosome* spp. responsible for AAT to obtain the conserved “proteome”. We departed from the five available sequences at TriTryDB r39 (*T. brucei* Lister427, *T. brucei* TREU927, *T. congolense* IL3000, *T. evansi* STIB805, and *T. vivax* Y486; ~50,000 CDS) and imposed the strictest conservancy possible: any residue position that was not identical in all five of them would be masked and thus not accounted for in the forthcoming steps (Figure 1). The reason to be so rigid was due to the limited number of genomes available. Upon applying CD-HIT we obtained about 30,000 protein clusters (Figure 1), amongst which we progressed only those antigens that were represented in all five departing proteomes. Therefore, we ran a custom script to identify those clusters with at least one representative protein sequence from each of those five proteomes and ended up with just 115 clusters that were aligned with MUSCLE [14]. Each of the 115 MSA was guided by a consensus protein (Figure 1). At this point, instead of having as reference proteins the unreal consensus sequences yielded by MUSCLE, we rather chose as reference the “real” sequence amongst all present in the MSA that most closely resembled that of the consensus. As a consequence, upon calculating the variability in the MSA with a rigid Shannon entropy *H* > 0.0 threshold, we obtained the invariant regions within those 115 potential antigens that were fully conserved in all five AAT-causing trypanosomes.

### 3.2. Prediction of B-Cell Epitopes

It was not possible to find experimentally validated AAT-conserved epitopes at the IEDB. We thus had to perform de novo predictions with immunoinformatic tools to potentially identify peptides of interest. These predictions were made on the invariant protein sequences common to the five proteomes.

We used Bepipred2.0 to predict B-cell epitopes that were at least 15 residues long. Before applying it to the invariant proteome file we manually selected those antigens in it that were expected to be surface-exposed by the annotation terms (Figure 1). Bepipred2.0 was run onto the FASTAs of those 21 “surface” antigens. Whenever there was a coincidence between the Bepipred2.0 prediction and the fact that such region of the antigen was fully conserved in the invariant proteome, then we had a predicted epitope to be further progressed. This yielded a total of 42 predicted epitopes (Additional Appendix A). Subsequent selection steps entailed prospecting the epitopes’ potential cross-reactivity to other protein sequences present in cows, as well as in sheep, goat, and pig proteomes. Four peptides were found to have >70% identity to either *B. taurus* or *B. indicus* or the described cow microbiome, so we discarded them from further progression. None of the peptides showed >70% identity to sheep, goat, or pig proteins. Moreover, since immunogenic B-cell epitopes are generally located at the surface exposed antigens, we ran subcellular localization tools to find out which antigens of origin of those epitopes were expected to be expressed at the surface of the trypanosomes. Any peptides that were predicted to have a subcellular localization to the mitochondrion with a “reliability class” (RC) 1–3, as determined by [21], or mapped to the internal regions of the surface-located antigens, were dismissed (shown in light grey in Additional Appendix A). Upon applying the cross-reactivity and subcellular localization criteria, we selected 31 predicted B-cell epitopes that originate from 11 different antigens (Table 1).

### 3.3. Prediction of T-Cell Epitopes

T-cell epitopes also had to be predicted. We used the IEDB Analysis Resource MHC-II and MHC-I binding tools to respectively predict CD4 and CD8 T-cell epitopes. The MHC-II binding tool at IEDB does not allow to specifically select class II bovine leukocyte antigens for the binding predictions so we used the available HLAs and the IEDB recommended set of tools [18]. Although the validated data are limited, MHC-II predictions using human alleles have been shown to be helpful towards the identification and validation of *Mycobacterium bovis* epitopes in cows [19]. We applied these MHC binding tools onto the invariant proteome of AAT-causing trypanosomes. No size limit was imposed to the predicted CD4 T-cell epitopes, whereas only nonameric epitopes were chosen for the prediction binding of CD8 T-cell epitopes. We respectively selected the 0.01% and 0.1% top-scoring epitopes amongst the CD4 T cell and CD8 T cell overall predictions. This way we ended up with 55 CD4 T-cell and 79 CD8 T-cell predicted epitopes, respectively (see Additional Appendix A). Upon applying the cross-reactivity threshold to proteins present in cows, as well as against the sheep, goat, and pig proteomes, we disregarded from further analysis 47 CD4 T-cell and 64 CD8 T-cell potential epitopes that had a >70% sequence identity to those. Therefore, we selected for inclusion in the vaccine ensemble 8 CD4 T-cell-predicted epitopes (Table 2) and 15 CD8 T-cell-predicted epitopes (Table 3). The CD4 T-cell peptides were predicted to bind to five different HLAs (Table 2), whereas the CD8 T-cell peptides were predicted to bind to 23 distinct BoLa alleles (Table 3). The percentage of identity (Id %) of the selected CD4 and CD8 T-cell epitopes to cow microbiome proteins as well as to protein sequences in sheep, goat and pig species can be checked in Additional Appendix A, respectively.

### 3.4. Design of a Genetic Vaccine Ensemble

The most economic scaffold to administer the vaccine ensemble would likely be genetic delivery of the epitopes as plasmid DNA (pDNA). We therefore designed a genetic construct that encompasses the 54 selected epitopes shaped as a “string-of-beads” [24]. Linkers of three glycine (GGG) residues were included to separate the epitopes and limit the formation of neo-epitopes [25,26]. It must be noted that besides experimentally validating the predicted epitopes, their optimum ordering within the ensemble and the spacers between them will have to be empirically determined. Nevertheless, in an attempt to anticipate a pDNA design we placed B-cell epitopes first right after a secretory signal to ensure their adequate processing to the surface, because B-cell epitopes are expected to be of utmost importance for eliciting a protective immunological response against AAT free-living trypanosomes [8] (Figure 2). Following the B-cell epitopes are the eight CD4 T-cell epitopes (Figure 2). These need to be physically connected to the B-cell epitope sequences in order to promote the correct processing and reactivity against the latter. Finally, the fifteen prioritized CD8 T-cell epitopes are placed towards the 3´-end of the “string-of-beads”, also separated by GGG spacers, for their optimal proteasome processing (Figure 2). 

Upstream of the epitope “string-of-beads” there is a cytomegalovirus (CMV) promoter sequence that governs its transcription, whereas at the string´s 3´-end there is a bovine growth hormone (BGH) poly-adenylation (polyA) signal to tag the growing mRNA molecule with a polyA tail. These two adjoining sequences are widely used within mammalian expression plasmids and they have been shown to work in the context of bovine cells, for instance [27,28]. 

Thinking of a delivery of the vaccine ensemble to cows, the full peptide sequence of the ensemble (Appendix A) was subjected to reverse translation using the *Bos taurus* code and its most likely abundant codons. As can be seen in Appendix A, the major codon dictionary used in *Bos taurus* differs in some residues with respect to that of AAT spp. However, plain reverse translation does not account for the optimal codon usage. This is a complex process where not only the codon usage is important, as it involves many other parameters, such as those facilitating transcription (GC content, CpG dinucleotides content, cryptic splicing sites, and occurrence of TATA boxes) or those increasing translation efficiency (codon usage bias, mRNA secondary structure, premature polyA sites, and RNA instability motifs). Thereby, we used an in vitro-translated mRNA (IVT) codon optimization tool [29] to optimize the reverse translated “string-of-beads” to be included in the pDNA. Alignment of the nucleotide sequence upon reverse translating the epitopes and linkers (with the *Bos taurus* codon usage; [30]) with the optimized nucleotide sequence for its expression in cows showed that both sequences share a high percentage of identity (77.9%), but many triplets were indeed optimized accordingly to the IVT sequence optimization tool [29] (Appendix A). 

Taking into account that the “string-of-beads” itself is 3498 nucleotides long, and that other regulatory sequences must go within the pDNA, such as the origin of replication (Ori) and the antibiotic resistance gene for propagation and selection (Kanamycin resistance in this case), we estimated a pDNA length of ~6.3 Kb. Such length can be now fully synthesized in vitro with a cow-optimized codon usage pattern.

### 3.5. 3D Modelling of the Peptide Vaccine Ensemble

The pDNA construct will be eventually expressed within cow cells to yield the poly-peptide epitopes. We used RaptorX to obtain a 3D model of what the vaccine ensemble would look alike upon being expressed. Notably, RaptorX also displays solvent accessibility (Acc) per position of all the residues within the submitted ensemble assigning a classification of buried (“B”), medium (“M”), or exposed (“E”) to each of these residues [23]. The retrieved 3D model of the AAT vaccine ensemble showed that regions with B-cell and CD4 T-cell epitopes mostly contained α-helixes as secondary structure, whereas the majority of the modelled β-sheets were in the CD8 T-cell epitopes C-term side of the ensemble (Figure 3). Moreover, large fragments of the B-cell epitope-containing N-term of the poly-peptide were modelled as disordered structures, which would theoretically make them more accessible for recognition by B-cell receptors and antibodies (Figure 3). Nonetheless, it must be noticed that the 3D model provided is just a model, and that the “combined protein” may just be one of the multiple possible combinations and not necessarily the best one.

If we take into account the Acc classification by RaptorX, it appeared that the highest percentage of “B” residues matched to CD4 and CD8 T-cell epitopes (35.2% and 33.3, respectively; Appendix A), whilst the percentage of B-cell epitopes “E” residues was 44.5% (Appendix A). The GGG spacers were mostly modelled as disordered structures (Figure 3), and thus 71.0% of them were tagged as “E” (Appendix A).

## 4. Discussion

In light of the lack of continuity in vector control programs and the increasing reports of drug resistance occurrence, vaccination appears to be a sought-after solution for the control of AAT impact. But the achievement of a functional vaccine product is in doubt due to the parasites ability to dampen any immune reactivity against them. Hence, alternative animal production methods, such as farming trypanotolerant breeds (e.g., N´Dama cows), have been proposed [4]. It must be noted though that these breeds are still sensitive to the disease in areas of high transmission rate, and there is the risk that trypanosome infections still pose an immunosuppressive toll, making cattle more vulnerable to other pathogens [4]. This is indeed an important obstacle to exploring trypanotolerance in most livestock production systems in sub-Saharan Africa, and enforces seeking a vaccine-based intervention. Hitherto, a certain level of protection against a challenge has been described in a variety of immunizing protocols in mouse models of trypanosome infection, including the inoculation of *T. congolense* recombinant transialidase [31] or immunization with other invariant antigens [9,10]. Furthermore, in a larger and more biologically relevant cow preclinical model of infection, inoculation of animals with irradiated *T. congolense* or *T. brucei* or with purified VSGs protected against the homologous challenge. Besides, trypanotolerant breeds have been described to have higher levels of anti-trypanosome immunoglobulins than trypanosusceptible cattle (reviewed in [11]). It should thus be possible to develop a vaccine for AAT. Nonetheless, since studies relying on classical approaches have mostly failed, innovative strategies might be required to overcome the challenge, e.g., a multi-antigenic epitope-based vaccine ensemble as the one proposed here. The increasing availability of genomic and other omics data on pathogens, allows researchers to use immune-informatics approaches towards a reverse vaccinology driven development [32]. Some of these tools are already being applied for the development of vaccines against cattle-relevant pathogens [33,34,35] or the search of new diagnostics for *T. vivax* [36].

We used publicly available and tailor-made computational methods to identify the epitopes presented in this article. Unfortunately, we did not find any validated AAT epitopes at the IEDB repository that complied with the conservation criteria, so we had to run de novo predictions. Therefore, the next step to take will be to experimentally validate that the predicted epitopes are presented and recognized in the cow immunological context, and that they are capable of eliciting immunogenic responses. Despite being based on predictions, the goodness of currently available tools for this purpose has been acknowledged by several authors and the use of prediction tools is seen as a great aid for the identification of new epitopes [19,36,37,38]. Since the protein space of trypanosomes, particularly *T. congolense*, *T. evansi*, and *T. vivax* remains largely unexplored, the majority of the antigens of origin of the predicted epitopes were either uncharacterized proteins or had low annotation scores inferred by homology. However, the antigens from where the B-cell epitope DSLCTLNMVPGVSVYGEKRVEVGATQ; CD4 T-cell epitopes AQIHDIYRMIPPLQVVLVSAT, DGRIGIILMDNITEVQSGQK, and AKNKFFYMYVQELNYLIRF; and CD8 T-cell epitopes MMHAYASRY, FGPWFVEPV, and EHNWMFAEI emerged, had *T. brucei brucei* homologues counting with medium to high annotation scores and experimental evidence of expression at either the transcriptional and/or protein level. The aforementioned B-cell and first CD4 T-cell epitope sequences, respectively matched to fibrillarin and a putative ATP-dependent RNA helicase. The other two CD4 T-cell epitopes matched to the *T. congolense* homologue of a well-characterized *T. b. brucei* mitochondrial ATP synthase subunit alpha (UniProt ref.: Q9GS23). This protein had experimental evidence provided by three different laboratories and it has been shown to be very important for the correct growth of the parasite [39]. Each of the three CD8 T-cell epitopes referred above was from a different intracellularly located antigen. The first two were respectively inferred by homology to *T. b. brucei* NADH dehydrogenase and proteasome subunit beta type-3; whereas the last one was an uncharacterized *T. b. brucei* protein with a >90% identity to the *T. equiperdum* chromosome passenger complex (CPC) protein. The uncharacterized *T. b. brucei* homologue and one of the ATP synthase subunit alpha had 3D crystal structures, and thus experimental evidence at the protein level.

Taking into account that (i) mixed infections with various trypanosome species have been reported [40]; (ii) the very limited resources available to develop a vaccine specific for each AAT-causing pathogen; and (iii) the fact that evolutionary conserved CDS regions of an organism are likely to have fundamental biological functions, sequence conservation was a key criteria to consider in the process of prediction of the epitopes that encompass the proposed vaccine ensemble. In fact, we asked for strict conservation between the protein coding regions of all five genomes from the AAT parasites, upon the hypothesis that immune responses elicited by invariant conserved antigens could protect against heterologous infections, too.

Trypanosome parasites responsible of AAT persist in the bloodstream thanks to a series of evolved mechanisms that allow them to escape innate immune controls and survive the successive waves of adaptive immune responses. At the initial phases of the infection, it has been described in mouse experimental models that *T. brucei* dampens macrophage pro-inflammatory responses, and both *T. brucei* and *T. congolense* can modulate in their favor the T-cell response functionality [8]. However, a series of studies in mice described that VSG-specific CD4^+^ T-cell responses can provide resistance to *T. brucei* infection [41]. Moreover, IFNγ secreted by Th1 cells was shown to activate macrophages, leading to the release of trypanosome killing factors and thus mediating resistance to *T. congolense* infection [42]. These Th responses were mainly mapped to epitopes in the variable regions of VSG antigens [41]. Nonetheless, the immunogenicity to CD4^+^ T cells specific of a VSG conserved peptide has also been described [43]. Thereby, the feasibility of the stimulation of peptide-specific immune responses in AAT infections with a series of MHC-II binding CD4 T-cell epitopes from conserved protein regions could support prophylactic cross-protection as well as the induction of functional memory cells.

In comparison to the suggested protective role of CD4^+^ T cells, CD8^+^ T cells were described to have a parasite-promoting effect in a mouse model of experimental *T. brucei* infection [44]. But CD8^+^ T cell depletion in trypanosusceptible Boran cattle naturally infected with *T. congolense* through tsetse bites did not lead to any effect in the parasitemia or anemia, which is the main determinant of survival [3]. Similarly, no immune-suppression caused by CD8^+^ T cells was observed in *T. evansi* experimentally infected sheep [45]. Thereby, the role played by CD8^+^ T cells in AAT infections is still obscure, and the potential usefulness of their inclusion in a vaccine product should not be disregarded. In fact, CD8^+^ T cells might promote certain level of protection yet to be discovered, since they are exacerbated during the natural infection process [46].

Nevertheless, being overall extracellular pathogens, the humoral response against AAT trypanosomes is likely to be the most important towards ultimately controlling the infections [47]. In order to counteract B-cell mediated responses these parasites pose complex and thick surface coats packed with homodimers of the VSG family, elevated on GPI anchors to conceal other “more sensible” antigens such as those within the ISG family [48]. Moreover, AAT parasites have evolved to antigenically switch their VSGs upon immune pressure in order to escape antibody-mediated destruction [48]. From the perspective of the modulation of host factors, mouse models of infection with *T. brucei* and *T. congolense* have shown that these trigger a polyclonal B-cell activation, which may contribute to take the immune attention away from relevant antigens, like VSGs (see [8] for a review). Besides, the two former trypanosome species and *T. vivax* have been described to disrupt B-cell lymphopoiesis in primary and secondary lymphoid organs, which affects the establishment of pools of memory B cells [8]. In sight of the major relevance of the antibody response to control parasitemia, a vaccine strategy that promotes a protective response against conserved regions of surface protein sequences may contribute to control AAT, especially if it is accompanied of CD4^+^ T cells inducing epitopes and the adjuvant used to exacerbate the exerted response biases it towards a Th1 profile. Whether such an adjuvant is sourced outside the vaccine construct or can be encoded within it or co-immunized in another construct [49], remains to be evaluated. Adjuvants will be required for effective vaccination and the latter choice seems more advisable as it will contribute to save costs.

Here we provide a tentative design for a pDNA ensemble, thinking of its likely reduced development and production costs in comparison to other delivery platforms. Following previously described constructs [27,28], we have included a CMV promoter signal and a BGH-pA, respectively upstream and downstream of the epitope “string-of-beads”. We did not include any extra immune enhancing signals in the Figure 2 map, but this type of booster could also be included within the pDNA as far as there is room for them. Nonetheless, it is advisable to keep the length of the pDNA under 7.5 kb longs, as it has been shown that the bigger the plasmid the less efficient will be the cell transfection, and there is an inverse correlation between the construct size and the final expression of the gene as the bigger the plasmid the less it can enter the nucleus [50]. 

At present there are five DNA vaccines already approved for veterinary use [51]. Remarkable advantages of a pDNA approach for the vaccination of large animals in tropical regions are that it can be easily produced at high quantities and does not require cold-chain. On the other side, it must be considered that effective immunization of large animals often fails [51], and adjuvants and/or prime-boost strategies should be pursued if a pure DNA vaccine is not potent enough. Immunization of large animals may face difficulties to reach sufficient concentration of antigens upon pDNA immunization, and thus this approach might work better if the animals are not too big, i.e. immunizing calves early in life with vaccine doses appropriately spaced [52]. 

Potency and longevity of vaccine-exerted immune response will be fundamental and will have to be determined experimentally. In this respect, the questioned poor immunogenicity concerns on pDNA vaccination might be circumvented by including codon optimization and administrating the construct in combination with the appropriate adjuvant. Besides, the pDNA delivery methodology and the prime–boost regime followed can be decisive [49]. Bearing in mind the results by Tabel et al. the route and dose of the antigen to be delivered will be key [53]. In this respect, electroporation could be used to enhance the exerted immune reactivity against pDNA immunization [28] and their field use considered [54]. 

## 5. Conclusions and Limitations

Despite there are two drugs available for the treatment of AAT the appearance of resistances is common due to poor attachment to treatment. Besides, the implementation and maintenance of vector control programs, crucial to control the disease, is irregularly performed as they are linked to economic and political stability, which may not be assured in the affected regions. For these reasons, the availability of a vaccine that could prevent the infection would mean a major breakthrough to limit the impact of AAT. Some efforts have been initiated but the biological complexity of the parasites and the very limited funding available for research on the subject has precluded its development.

We propose an alternative vaccine approach based on epitopes from a range of different parasite antigens to cope with the parasites antigenic complexity. The epitopes were prioritized in terms of their absolute conservation among all the sequenced genomes of the distinct AAT-causing spp. The main rationale, besides taking advantage of the biological relevance of highly conserved protein motifs for the parasites´ biology, is to ultimately provide a single vaccine ensemble to tackle all ATT-causing trypanosomes. Progressing a single vaccine product would contribute to save developmental costs. Moreover, we propose a pDNA vaccine construct to carry the epitope-based “string-of-beads” because it could be synthetized at a fraction of the cost of producing recombinant protein subunits or virus vectored vaccines. Remarkably, it is also a platform capable of yielding the high amounts of product needed to immunize large animals, which can be lyophilized and suitably used in tropical regions. All epitopes in the ensemble were prioritized in agreement with their lack of identity to any protein sequences present in cows, and in the proteomes of sheep, goat, and pig species, so as to avoid the appearance of cross-reactivity reactions, thus favoring the safety of the vaccine but also likely promoting its immunogenicity.

A major limitation of the study is that we did not find experimentally validated conserved AAT epitopes in the IEDB repository database, so we had to rely on de novo predictions of epitopes derived from the computed invariant proteome file. We applied a very conservative threshold and only top qualifying predicted epitopes were selected for progression. Nonetheless, whether they are presented and recognized in the course of an infection will need to be experimentally demonstrated. Appropriate processing is a key feature towards the immunogenicity of the epitopes, and this will have to be thoroughly studied in relation to the delivery of the epitopes in the form of a genetic construct. Furthermore, their optimum ordering and spacing within the ensemble will also require to be adjusted experimentally as well as the selection of the most adequate adjuvant. Thereby, the pDNA ensemble provided here must just be considered as a theoretical scheme, subject to any required reformulations upon validation of the predicted epitopes and the study of their immunogenicity and capacity to induce protective immune responses. Other acknowledged limitations of the reverse vaccinology strategy we have followed are the inability to provide glycan-based epitopes and the absence of conformational B-cell epitopes, since these cannot be isolated from their context and thus we only focused on linear ones.

In an attempt to anticipate a vaccine candidate, we placed the B-cell epitope component first as it is expected that an antibody-mediated response will play a major role in controlling free-living AAT infections. In the construct these are immediately followed by CD 4 T-cell epitopes, which will be fundamental for the correct response to all the former, but also because there is evidence of the importance of CD4^+^ T cells against some AAT-causing trypanosomes. Finally, since the functionality of CD8^+^ T cells is still under debate, we placed them at the end of the string but decided to keep them in the proposed ensemble, as CD8^+^ T cells may play a role in eliminating infected macrophages and, therefore, freeing the suppressive control they have on AAT immunity. 

This work lies at a very preliminary stage of the long vaccine development path. Notwithstanding, one of the advantages of an in silico vaccinology approach is that it can contribute to saving time and resources. Hopefully, the epitopes here selected can reach experimental studies in the near future and prove they can indeed elicit immunogenic responses against infections with AAT trypanosomes.

## Figures and Tables

**Figure 1 vaccines-08-00130-f001:**
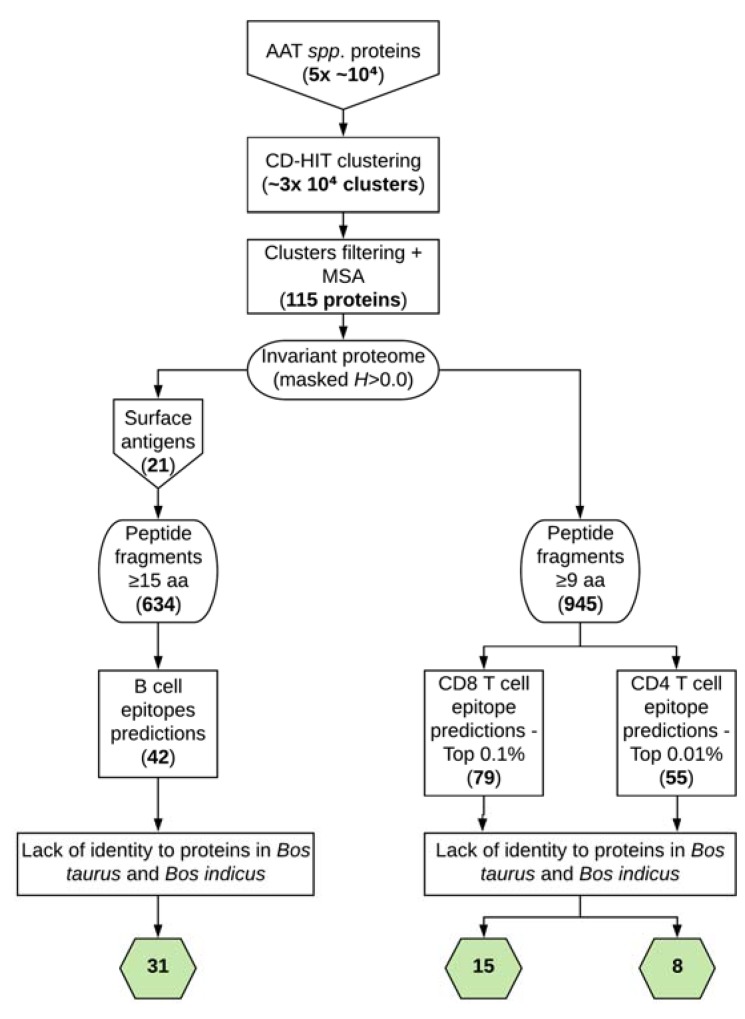
Flowchart summary of the epitope identification strategy. Larger boxes show the steps and the hexagons the numbers of finally selected epitopes. This figure was made at https://www.lucidchart.com.

**Figure 2 vaccines-08-00130-f002:**
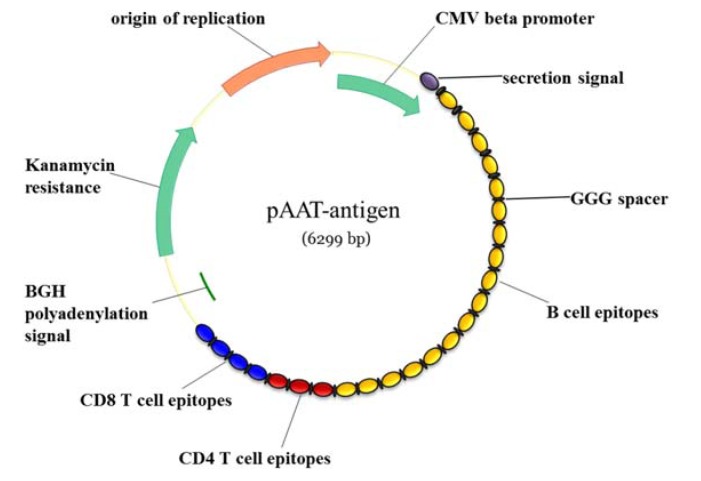
Map of the proposed pDNA vaccine construct.

**Figure 3 vaccines-08-00130-f003:**
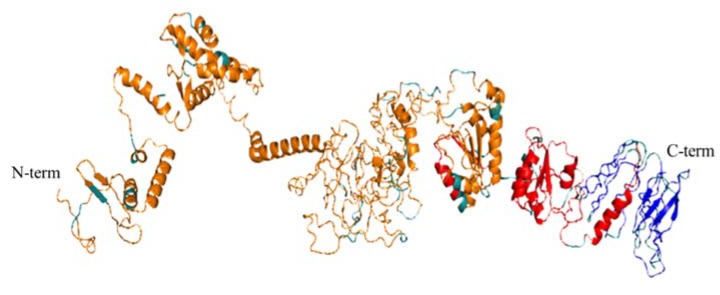
3D model of the vaccine ensemble generated by RaptorX, respectively encompassing from the N-term to the C-term: B-cell epitopes (orange), CD4 T-cell epitopes (red), and CD8 T-cell epitopes (blue). GGG linkers are shown in gray.

**Table 1 vaccines-08-00130-t001:** List of the B-cell epitopes predicted in the invariant AAT-spp. proteome selected for the ensemble.

Epitope	Length	Antigen ^1^	Annotation ^1^	TargetP ^2^	*Bos taurus* Hit ^3^	Id (%) ^4^	*Bos indicus* Hit ^3^	Id (%) ^4^
MQYGSTPKDIRYGIE	15	TcIL3000.11.15250.1	chaperonin HSP60, mitochondrial precursor	Other (5)	NP_001029859.1	28.6	XP_019842974.1	31.6
GFTSPYFVTNTKSQKC	16	TcIL3000.11.15250.1	chaperonin HSP60, mitochondrial precursor	Other (5)	AAI02078.1	56.3	XP_019829938.1	56.3
VFTGAQMISEDLGLSLDQS	19	TcIL3000.11.15250.1	chaperonin HSP60, mitochondrial precursor	Other (5)	XP_024847167.1	23.3	XP_019820980.1	47.6
TISRDECILMEGGGSAIAVE ERVQMIRDMIAAEDHEYNRERL	42	TcIL3000.11.15250.1	chaperonin HSP60, mitochondrial precursor	Other (5)	XP_024847167.1	23.3	XP_019820980.1	47.6
RIEDRGLEDKEKREGLN	17	TcIL3000.11.15250.1	chaperonin HSP60, mitochondrial precursor	Other (5)	XP_024847167.1	23.3	XP_019811187.1	47.1
MRKRVNESQAPLPAL	15	TcIL3000.11.15250.1	chaperonin HSP60, mitochondrial precursor	Other (5)	NP_001029859.1	28.6	XP_019842974.1	31.6
KCIYYVTGDSKKKLETSPFIEQAKRRG	27	TcIL3000_0_26140.1	Heat shock protein 83, putative	Other (1)	NP_001073105.1	48.1	CCA61548.1	48.1
EYIPRAFPVKSTTGL	15	TcIL3000_10_3450.1	hypothetical protein, conserved	M (4)	XP_024851069.1	65.4	XP_019820927.1	65.4
ECQDDWCTLEKRYFW	15	TcIL3000_10_3450.1	hypothetical protein, conserved	M (4)	XP_024851069.1	65.4	XP_019820927.1	65.4
LETAQKWNEWRFWELDPEQVKKVAREDQNIGR EGLGYNSPWEQVVREDFDKRKKALTQEEMAE LKRMDTERMARETAAYKERKDRIRDDLEKARG	95	TcIL3000_10_3450.1	hypothetical protein, conserved	M (4)	NP_001029859.1	28.6	XP_019842974.1	31.6
KFLNKRFAVDKDLQRMQPGKRYSGKTAD	28	TcIL3000_10_3450.1	hypothetical protein, conserved	M (4)	NP_001029859.1	28.6	XP_019834515.1	28.6
PFGRFAHTPTVLPDSSIDLSYEVPWW	26	TcIL3000_10_510.1	hypothetical protein, conserved	Other (3)	NP_001029859.1	28.6	XP_019842974.1	31.6
DSLCTLNMVPGVSVYGEKRVEVGATQ	26	TcIL3000_10_6460.1	fibrillarin, putative	Other (3)	XP_024851069.1	65.4	XP_019820927.1	65.4
TGQFEKAGEQKEREGKH	17	TcIL3000_10_970.1	intraflagellar transport protein 172, putative	M (5)	XP_024847167.1	23.3	XP_019820980.1	47.6
DFDCTDFPKKYPMPKSSS	18	TcIL3000_10_970.1	intraflagellar transport protein 172, putative	M (5)	-	-	-	-
ADDGYVGYDSVPFHRYNR	18	TcIL3000_7_2400.1	hypothetical protein, conserved	M (4)	-	-	-	-
LGCATCKMPNDINEA	15	TcIL3000_7_510.1	hypothetical protein, conserved	Other (3)	NP_001029859.1	28.6	XP_019842974.1	31.6
IIKGCNVFELDGSMSDVHQSI	21	TcIL3000_7_510.1	hypothetical protein, conserved	Other (3)	AAI02078.1	56.3	XP_019829938.1	56.3
TSAAGLSPDWLEAFFTNVAYNT	22	TcIL3000_7_510.1	hypothetical protein, conserved	Other (3)	XP_024847167.1	23.3	XP_019820980.1	47.6
KSLRMNFFTVCERCVLKEM	19	TcIL3000_7_510.1	hypothetical protein, conserved	Other (3)	NP_001029859.1	28.6	XP_019842974.1	31.6
EKGPSTLPPLEHLFVASVYLAAQRQFTNLFFF	32	TcIL3000_7_510.1	hypothetical protein, conserved	Other (3)	XP_024851069.1	65.4	XP_019820927.1	65.4
DLKEWCKQYEQLTNDMLRLR	20	TcIL3000_7_510.1	hypothetical protein, conserved	Other (3)	-	-	-	-
ELPGAITHKSILELR	15	TcIL3000_8_4670.1	hypothetical protein, conserved	Other (2)	XP_024851069.1	65.4	XP_019820927.1	65.4
QSQQQLLASTRGGMPAR	17	TcIL3000_8_4670.1	hypothetical protein, conserved	Other (2)	XP_024847167.1	23.3	XP_019816904.1	23.3
SGVRHLRMAGDGTVGQN	17	TcIL3000_9_6470.1	hypothetical protein, conserved	M (5)	XP_024847167.1	23.3	XP_019820980.1	47.6
DWLERQFIDNCATPERDP	18	TcIL3000_9_6470.1	hypothetical protein, conserved	M (5)	XP_024851069.1	65.4	XP_019820927.1	65.4
LLDWEDFGIPREDLYR	16	TcIL3000_9_6470.1	hypothetical protein, conserved	M (5)	NP_001029859.1	28.6	XP_019842974.1	31.6
QQTRFDNTTEEKLRSL TYTQTDKTVDEYYG	30	TevSTIB805.5.5170-t26_1	hypothetical protein, conserved	S (3)	XP_024847167.1	23.3	XP_019816904.1	23.3
FPDPPFNFSAVVPER	15	TevSTIB805.5.5170-t26_1	hypothetical protein, conserved	S (3)	XP_024851069.1	65.4	XP_019820927.1	65.4
SGVKANDTGVGPNTTNTAGGA	21	TevSTIB805.5.5170-t26_1	hypothetical protein, conserved	S (3)	XP_024847167.1	23.3	XP_019820980.1	47.6
CVDIIDWRDLDEMLNNRTDEVVEKSL	26	TevSTIB805.5.5170-t26_1	hypothetical protein, conserved	S (3)	-	-	-	-

^1^ Antigen name and annotation are those reported in the parasite proteome of origin from TriTrypDB [12]. ^2^ Subcellular localization of antigens predicted with TargetP1.1 [21]: M, mitochondrial; S, secreted; number in brackets stands for the prediction “Reliability class” (RC), which ranges from 1 to 5 where 1 indicates the strongest prediction. ^3^
*Bos taurus* and *Bos indicus* non-redundant protein collection protein sequences are identified by their accession reference number at NCBI. ^4^ Percentage of identity (number of identical residues per queried length) of each epitope to its corresponding blasted hit (NCBI Accession name). The “-“ means there is no hit. The Id % to the cow microbiome proteins of the B-cell epitopes selected can be checked in Additional Appendix A.

**Table 2 vaccines-08-00130-t002:** List of the CD4 T-cell epitopes predicted in the invariant AAT-spp. proteome that were selected for the ensemble.

Epitope	Antigen ^1^	Annotation ^1^	HLAs (human) ^2^	*Bos taurus* Hit ^3^	Id (%) ^4^	*Bos indicus* Hit^3^	Id (%) ^4^
AQIHDIYRMIPPLQVVLVSAT	TcIL3000.11.9080.1	ATP-dependent RNA helicase FAL1, putative	HLA-DRB3*02:02	NP_001039653	66.7	XP_019837752	66.7
FKAQIHDIYRMIPPLQVVLVS	TcIL3000.11.9080.1	ATP-dependent RNA helicase FAL1, putative	HLA-DRB1*09:01	NP_001039653	66.7	XP_019837752	66.7
GLVIERRLSDKHFVF	TcIL3000_0_22360.1	NADH-dependent fumarate reductase	HLA-DRB3*01:01	3FE5_A	46.7	XP_019825021	46.7
RINLVVQFDMASDADSYLHRV	TcIL3000_10_400.1	ATP-dependent RNA helicase SUB2, putative	HLA-DRB3*01:01	NP_001029924	57.1	XP_019819542	57.1
DGRIGIILMDNITEVQSGQK	TcIL3000_7_6050.1	ATP synthase F1, alpha subunit, putative	HLA-DRB3*01:01	XP_005203166	40.0	XP_019836600	40.0
AKNKFFYMYVQELNYLIRF	TcIL3000_7_6050.1	ATP synthase F1, alpha subunit, putative	HLA-DPA1*02:01 _DPB1*14:01	XP_005205520	36.8	XP_019814319	36.8
ASYANVWEMDDPYRFLQTEQD	TcIL3000_9_4570.1	Chromosome passenger complex (CPC) protein INCENP N terminal, putative	HLA-DRB3*01:01	DAA31134	38.1	XP_019813076	38.1
DYTNRIIRQMLHNVAALSCNK	TevSTIB805.10.4720-t26_1	Pumillo RNA binding protein PUF1	HLA-DRB4*01:01	XP_002697522	38.1	XP_019840994	38.1

^1^ Antigen name and annotation are those reported in the parasite proteome of origin from TriTrypDB [12]. ^2^ HLA alleles to which the IEDB tool “MHC-II Binding Prediction” (http://tools.iedb.org/mhcii/) reported a high binding prediction. ^3^
*Bos taurus* and *Bos indicus* non-redundant protein collection protein sequences are identified by their accession reference number at NCBI. ^4^ Percentage of identity (number of identical residues per queried length) of each epitope to its corresponding blasted *Bos taurus* and *Bos indicus* hit.

**Table 3 vaccines-08-00130-t003:** List of the CD8 T-cell epitopes predicted in the invariant AAT-spp. proteome that were selected for the ensemble.

Epitope	Antigen ^1^	Annotation ^1^	BoLA ^2^	*Bos taurus* Hit ^3^	Id (%) ^4^	*Bos indicus* Hit ^3^	Id (%) ^4^
MMHAYASRY	TcIL3000.11.1250.1	NADH dehydrogenase [ubiquinone] iron -sulfur protein 7, mitochondrial	BoLA-N:05501, BoLA-N:02401	DAA15150.1	66.7	XP_019842744.1	66.7
YPAVRVYPV	TcIL3000.11.9220.1	NADH dehydrogenase subunit NB6M, putative	BoLA-N:00501	DAA32829.1	66.7	XP_019825300.1	66.7
FGPWFVEPV	TcIL3000_0_05230.1	proteasome beta 3 subunit, putative	BoLA-N:05301, BoLA-N:00402, BoLA-N:00401, BoLA-N:00102, BoLA-N:00103, BoLA-N:00101	NP_789861.1	55.6	XP_019823249.1	55.6
VTPSIHYTM	TcIL3000_0_22360.1	NADH-dependent fumarate reductase	BoLA-N:05301, BoLA-N:03801, BoLA-N:00402, BoLA-N:00401	XP_024843166.1	66.7	XP_019834535.1	66.7
ATFEVFHTI	TcIL3000_10_13200.1	methyltransferase domain containing protein, putative	BoLA-N:05401	XP_005225549.2	66.7	XP_019808611.1	66.7
LQYHKYGCL	TcIL3000_10_3580.1	ubiquinol-cytochrome c reductase complex 14kD subunit, putative	BoLA-HD6	NP_001179015.1	66.7	XP_019841558.1	66.7
YGPSWHCVM	TcIL3000_10_3900.1	dynein light chain, putative	BoLA-N:05301, BoLA-N:04001, BoLA-N:00402, BoLA-N:00401, BoLA-N:00102, BoLA-N:00103, BoLA-N:00101	XP_015328853.2	55.6	XP_019837337.1	55.6
DMYLHQHEF	TcIL3000_10_970.1	intraflagellar transport protein 172, putative	BoLA-NC1:00101, BoLA-NC1:00201, BoLA-NC1:00301	XP_002693758.2	55.6	XP_019830009.1	55.6
EEMKYVAGL	TcIL3000_10_970.1	intraflagellar transport protein 172, putative	BoLA-N:04101	NP_001192842.1	66.7	XP_019822633.1	66.7
RKYETTWEM	TcIL3000_6_4030.1	hypothetical protein, conserved	BoLA-D18.4	AAI42017.1	55.6	XP_019813499.1	55.6
EHNWMFAEI	TcIL3000_9_4570.1	Chromosome passenger complex (CPC) protein INCENP N terminal, putative	BoLA-N:05001	XP_024855981.1	55.6	XP_019826963.1	55.6
SEMRAWYWK	TcIL3000_9_4570.1	Chromosome passenger complex (CPC) protein INCENP N terminal, putative	BoLA-N:04201	NP_001095539.1	55.6	XP_019817650.1	66.7
AMWSQDSPY	TcIL3000_9_6470.1	hypothetical protein, conserved	BoLA-N:00901	NP_001193092.1	66.7	XP_019825232.1	66.7
ARYEYFLAY	TevSTIB805.10.7310-t26_1	Sterol methyltransferase, putative	BoLA-N:02201	XP_005212827.1	55.6	XP_019825865.1	55.6
THETHSFLY	TvY486_1100540:mRNA	dynein light chain, putative	BoLA-N:04801	XP_024852897.1	66.7	XP_019822516.1	66.7

^1^ Antigen name and annotation are those reported in the parasite proteome of origin from TriTrypDB [12]. ^2^ BoLa alleles to which the IEDB “MHC-I Binding Prediction” (http://tools.iedb.org/mhci/) reported a high binding prediction. ^3^
*Bos taurus* and *Bos indicus* non-redundant protein collection protein sequences are identified by their accession reference number at NCBI. ^4^ Percentage of identity (number of identical residues per queried length) of each epitope to its corresponding blasted *Bos taurus* and *Bos indicus* hit.

## Data Availability

The data used to support the findings of this study are included within the article or within the Appendix A. The scripts developed are available from the corresponding author upon request.

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
