# Peer review of "Design of an Epitope-Based Vaccine Ensemble for Animal Trypanosomiasis by Computational Methods"

_vaccines, 2020, doi:10.3390/vaccines8010130_

Round 1

Reviewer 1 Report

The manuscript by Michel-Todó et al. describes a study where they have designed a multi-epitope-based vaccine ensemble for animal trypanosomiasis using various computational models. This is an interesting study and described methods to identify vaccine candidates using a revers vaccinology approach. However, it remains to be seen if this vaccine construct elicits any in vitro immunological activity and in vivo humoral or cell-mediated immune response with or without an adjuvant.

General comments:

The manuscript is written and presented very well.

Since the manuscript involves development of a vaccine construct and not the vaccine, therefore the title of the manuscript should be ‘Design of an epitope-based vaccine ensemble or construct for animal trypanosomiasis by computational methods.’

Authors have suggested a pDNA based vaccine delivery strategy. However, the DNA is inherently unstable and challenging to deliver inside the host cell nucleus. Alternatively, mRNA can be used to express the protein of interest in the host cells using viral and non-viral delivery systems or mRNA modifications. mRNA are showing great promise and should be discussed as an alternative to pDNA vaccine strategy.

Author Response

Review Report Form – R1

The manuscript by Michel-Todó et al. describes a study where they have designed a
multi-epitope-based vaccine ensemble for animal trypanosomiasis using various
computational models. This is an interesting study and described methods to identify
vaccine candidates using a revers vaccinology approach. However, it remains to be seen
if this vaccine construct elicits any in vitro immunological activity and in vivo humoral
or cell-mediated immune response with or without an adjuvant.

General comments:

The manuscript is written and presented very well.

Thank you very much.

Since the manuscript involves development of a vaccine construct and not the vaccine,
therefore the title of the manuscript should be ‘Design of an epitope-based vaccine
ensemble or construct for animal trypanosomiasis by computational methods.’

We have now modified the title and short title to answer the first general comment by
R1.

Authors have suggested a pDNA based vaccine delivery strategy. However, the DNA is
inherently unstable and challenging to deliver inside the host cell nucleus. Alternatively,
mRNA can be used to express the protein of interest in the host cells using viral and
non-viral delivery systems or mRNA modifications. mRNA are showing great promise
and should be discussed as an alternative to pDNA vaccine strategy.

In relation to this second comment, we thank R1 for bringing onto the table the
possibility to use mRNA as potential molecular backbone of the vaccine ensemble. This
feature was in fact discussed between the manuscript authors during the drafting of the
first version, particularly bearing in mind the context where vaccine would be deployed
(tropical climate in low resources settings) and that large animals were the target. Upon
considering several arguments and revising the literature in the subject we agreed on
suggesting that a pDNA construct would be more desirable.

In relation to the DNA molecules instability, actually it is largely documented in the
literature that DNA (and specially pDNA) is much more stable than RNA, particularly
in vivo where RNAses are ubiquitous. In fact, pDNA was described to be detected in a
non-integrated form in muscle up to six months following injection (Ledwith BJ et al.
Plasmid DNA vaccines: investigation of integration into host cellular DNA following
intramuscular injection in mice. Intervirology 2000, 43, 258–272).

Although promising, the mRNA technology did not yet come out of clinical trials, were
only 10 to 15 studies are registered for mRNA administration, all in phase 1, and mainly
by three companies (Moderna Therapeutics, BioNTech Inc. and Curevac Inc.). In the
field of infectious diseases, no outcome of mRNA-based clinical trials is yet available,

except that of 2 HIV studies: one was terminated because of lack of immunogenicity
(NCT02888756), and the other one showed promising phase 1 results (NCT02413645).

Please, note that most of the clinical trials using mRNA aim at ex-vivo transfecting
dendritic cells before their re-administration and are not taken into account here.

On the contrary, the use of pDNA has been extensively studied for more than 20 years,
and no major drawbacks were identified. After showing great promise in the beginning,
it still took about 15 years for the development of veterinary products, and no human
one is still available. Since five veterinary vaccines based on pDNA technology are
already marketed, it is reasonable to think that it could be possible to ever license a
pDNA vaccine for AAT, providing the efficacy is good enough.

The advantages and disadvantages of mRNA and pDNA vaccines have been extensively
discussed recently (Liu MA. A comparison of plasmid DNA and mRNA as vaccine
technologies. Vaccines (Basel) 2019; 7(2). pii: E37). As stated there, the mRNA field is
currently following the same path as the pDNA field 20 years ago as far as development
is concerned, and some issues still need to be solved. For example, the real efficacy in
large animals to raise a relevant immune response, the possible toxicity of mRNA
molecules containing unnatural modified nucleosides, the delivery routes… As a result,
we consider that the choice for the potential development of a vaccine product to be
used in low-income countries against AAT infections should be a pDNA vaccine.

Reviewer 2 Report

In this manuscript, Todo et al predicted B/T cell epitope for four species of Trypanosoma by using published/online servers. The study was described clearly and the results are reasonable based on the observations. However, there are some concerns need to be addressed before publication.

Major Concerns:

1, As discussed in the manuscript, the authors did not test any of the predicted epitopes by in vitro or in vivo models, which is very import for prediction study.

2, there are many combinations of the expression cassette for the combined epitopes. It is meaningless to predict the structure of the “combined protein”, not only because it may not be the best one, but also due to the lack of solid structural template.

3, the authors also need to pinpoint how many successful cases (in phase I/II/III trial, or licensed) have been attained in the field of epitope vaccine.

Author Response

Review Report Form – R2

Comments and Suggestions for Authors

In this manuscript, Todo et al predicted B/T cell epitope for four species of
Trypanosoma by using published/online servers. The study was described clearly and
the results are reasonable based on the observations. However, there are some concerns
need to be addressed before publication.

Major Concerns:

1. As discussed in the manuscript, the authors did not test any of the predicted epitopes
by in vitro or in vivo models, which is very import for prediction study.

We understand this issue and highlight it as a limitation of the study in several parts of
the text: page 10, lines 222-224; p12, lines 303-307; p14, lines 430-438. We further
agree that the immediate next steps within this research line would be to experimentally
validate the predicted epitopes by means of in vitro and/or in vivo experiments as R2
comments. Unfortunately at present we do not have the means to do so.

In relation with the inclusion of predicted epitopes in the ensemble we can also say that,
as part of the conceived set of events of the work flow, we first investigated whether
there were already validated epitopes at IEDB repository which we could rely on for the
design of the vaccine construct. We tried to follow a legacy experimentation approach,

same as Quinzo MJ et al. recently published work on the computational design of a
vaccine for human cytomegalovirus infection
(https://www.ncbi.nlm.nih.gov/pubmed/31823715). However, there was a very limited
number of experimentally validated epitopes at IEDB from AAT-causing trypanosomes,
and with the conservation criteria we applied in this work, we could not find any at all.
Something similar happened to us in our recently published manuscript about in silico
design of a vaccine construct for the closely related Trypanosoma cruzi parasite
(Michel-Todó et al., Front Immunol 2019). There, from the 30 final epitope sequences
selected for further progression, we could only find one validated epitope (at IEDB).
Certainly, in relation to the pathogens responsible for these neglected infections there is
yet a lot of work to do, and despite the advancements made in the last few decades their
protein space remains largely unexplored as it is the immunological study of many of
their antigens.

2. There are many combinations of the expression cassette for the combined epitopes. It
is meaningless to predict the structure of the “combined protein”, not only because it
may not be the best one, but also due to the lack of solid structural template.

We thank R2 for this comment and align to it. Indeed the number of possible different
combinations of epitopes within the potential ultimate cassette is huge and it could most
probably not be reasonably addressed experimentally. Thus we tried to prioritize the
positioning of the distinct types of epitopes in relation to their assigned role in the
immune control of free-living trypanosomes infection in the bloodstream.

On the other hand, the rationale to provide a model of the potential 3D structure, in
agreement with the previously published work on T. cruzi (Michel-Todó et al., Front
Immunol 2019), was to potentially “detect” the space localization of cleavage sites in
between the suggested epitopes as well as whether the B cell epitopes included could or
could not be accessible for antibody recognition. This said, we do agree with R2
concerns on the fact that the 3D model provided is just that, a model, and that the
“combined protein” may just be one of the multiple possible combinations and not
necessarily the best one. It could probably happen that the construct will not fold in a
stable manner and the model would just represent one of the many possible
conformations possible. A new sentence has now been included in the text on this
matter (page 11, line 269-270).

3. The authors also need to pinpoint how many successful cases (in phase I/II/III trial, or
licensed) have been attained in the field of epitope vaccine.

In relation to this third concern by R2, although the literature regarding multi-epitope
vaccine is quite abundant (about 400 publications) very few of them have reached the
clinical stage yet. Indeed, most current work is aimed at identifying relevant poly-
epitopes, taking profit of the recent progresses made in computational tools. When
searching the clinical trial website (https://clinicaltrials.gov/ct2/home), less than ten
studies are really related to multi-epitope vaccines testing. Among them, 5 are dedicated
to the immunotherapy of various cancers using multi-epitope peptides (NCT00580060,
NCT02362451, NCT04144023, NCT03012100, NCT02764333). Whereas just three are
pDNA studies: two for cancer immunotherapy (NCT01322802, NCT01138410), and the

other for Plasmodium falciparum induced malaria (NCT01169077). All these trials are
either phase I or I/II, and just two have published results. The malaria NCT01169077
trial failed to show induction of an immune respose, while the NCT01138410 against
melanoma showed induction of a potent T cell response.

However, we must say that the issue with epitope-based vaccines is that they are the last
resource, which means that they are applied to those cases where the rest of possible
vaccination strategies have already failed. This would be the case of vaccination against
complex large viruses or parasite pathogens. For the latter, where the lack of resources
is indeed a major limitation for the development of vaccine products despite they could
mean major breakthrough in the management of many of these infections, a strategy
such as the computationally fed multi-epitope multi-antigen could be the solution.

Reviewer 3 Report

The manuscript entitled “Design of an epitope-based vaccine for animal Trypanosomiasis by computational methods” by Lucas Michel-Todó and co-workers reports a reverse vaccinology approach to developing plasmid-based vaccines for African animal trypanosomiasis. The authors used bioinformatics approaches to select 31 B cell, 8 CD4 T cell and 15 CD8 T cell epitope sequences from 30 distinct antigens as candidate antigens with strict sequence conservation between five available Trypanosome spp proteasome sequences. The authors also proposed to use plasmid-based vaccination approaches to deliver all the candidate antigens (as epitopes) to animals, and provided a scaffold for the plasmid. The bioinformatics analysis is well designed and present in a clear logical. I have the following concerns:

Major concerns

1). The approaches and conclusions the authors made in this manuscript are all based on a simple assumption, which is that the antigens of Trypanosomiasis disease must have conserved peptide sequences, and also the antigens are effective as epitopes. This assumption has no experimental data to support, so it makes the entire basis of the manuscript not solid. The well studied VSG based immunization approach may provide a clue that epitopes work as efficient as whole biological complex, however, there is no evidence the antigens must have strictly conserved peptide sequences. Considering the fact that different pathogenic strains usually show variable infection severity to the same host species and the presence of nonpathogenic trypanosomes, it is very suspicious that the antigens have strictly conserved peptide sequences.

2). The authors only compared the candidate epitopes sequences with the proteasome of cow and the corresponding microbiome to avoid potential cross-reactivity of those epitopes. However, the hosts of Trypanosome are not limited to cows, but also included sheep, goats, horses, and pigs. From a bioinformatics perspective, the authors should also include at least one another host species for eliminating potential cross-reactivity.

Minor concerns:

1). Among the 30 antigens the authors proposed in the manuscript, do any of those have been identified experimentally as the antigen for the infection. If any of those does, the authors should include the evidence in the manuscript to make the analysis more applicable.

Author Response

Review Report Form – R3

Comments and Suggestions for Authors

The manuscript entitled “Design of an epitope-based vaccine for animal
Trypanosomiasis by computational methods” by Lucas Michel-Todó and co-workers
reports a reverse vaccinology approach to developing plasmid-based vaccines for
African animal trypanosomiasis. The authors used bioinformatics approaches to select
31 B cell, 8 CD4 T cell and 15 CD8 T cell epitope sequences from 30 distinct antigens
as candidate antigens with strict sequence conservation between five available
Trypanosome spp proteasome sequences. The authors also proposed to use plasmid-
based vaccination approaches to deliver all the candidate antigens (as epitopes) to
animals, and provided a scaffold for the plasmid. The bioinformatics analysis is well
designed and present in a clear logical. I have the following concerns:

Major concerns

1) The approaches and conclusions the authors made in this manuscript are all based on
a simple assumption, which is that the antigens of Trypanosomiasis disease must have
conserved peptide sequences, and also the antigens are effective as epitopes. This
assumption has no experimental data to support, so it makes the entire basis of the
manuscript not solid. The well-studied VSG based immunization approach may provide
a clue that epitopes work as efficient as whole biological complex, however, there is no
evidence the antigens must have strictly conserved peptide sequences. Considering the
fact that different pathogenic strains usually show variable infection severity to the same
host species and the presence of nonpathogenic trypanosomes, it is very suspicious that
the antigens have strictly conserved peptide sequences.

We understand the concern formulated by R3, but on the other hand it could also be
assumed that fully evolutionary conserved regions between four distinct parasite species
may represent key biological standpoints for the living of these parasites and thus
potentially highlight very interesting targets for therapeutic intervention, either in the
form of chemotherapeutics or vaccination. Moreover, when dealing with neglected
infections it must be born in mind that the available funds to pursue research are almost

always very limited and thus the design and development of a single therapeutic product
that could counteract the deleterious impact of four distinct pathogenic species would
surely be an absolute breakthrough. Perhaps this departing hypothesis is too ambitious,
but it has been devised thinking of the long way yet to go, and considering the notable
cost limitations that will have to be faced.

It is also worth noting that conserved epitopes among different pathogens do exist. We
have actually pioneered a new strategy in vaccine design that lies in the selection of
experimental epitopes that are conserved (Quinzo et al., BMC Bioinformatics 2019).
Such conserved epitopes can therefore be seen by the immune system, and although
they are generally subdominant, vaccination can turn them into immune-dominant
sequences.

2) The authors only compared the candidate epitopes sequences with the proteasome of
cow and the corresponding microbiome to avoid potential cross-reactivity of those
epitopes. However, the hosts of Trypanosome are not limited to cows, but also included
sheep, goats, horses, and pigs. From a bioinformatics perspective, the authors should
also include at least one another host species for eliminating potential cross-reactivity.

We agree that there are other breeding species affected by animal trypanosomiasis of
major economic relevance. From this perspective, sheep, goats and pigs would probably
have the most relevance. Thereby we re-ran the cross-reactivity identifying scripts
against the reference proteomes of those three species and the results are now included
in the Additional Table S1 file. New lines have now been included in the text referring
to this new feature (page 1, line 26; p2, line 71; p3, lines 131-133; p5, lines 174-179;
p8, line 202;…), which also directs readers towards the Additional Table where the
dataset is.

Minor concerns:

1) Among the 30 antigens the authors proposed in the manuscript, do any of those have
been identified experimentally as the antigen for the infection. If any of those does, the
authors should include the evidence in the manuscript to make the analysis more
applicable.

We have carefully revised the annotation and characteristics of the 30 antigens of origin
of the epitopes predicted (in UniProt and TriTrypDB). Since the protein space of
trypanosomes, particularly T. congolense, T. evansi and T. vivax remains largely
unexplored, the majority of those antigens were either uncharacterized proteins or had
low annotation scores inferred by homology. Nonetheless, yet a few of the antigens
were >90% homologues to proteins with experimental evidence at the transcriptional
and/or protein level. The most interesting being TcIL3000_7_6050.1 from which two
CD4 T cell epitopes originated: DGRIGIILMDNITEVQSGQK and
AKNKFFYMYVQELNYLIRF. That antigen was homologue to T. brucei brucei
UniProt ref. Q9GS23, shown to be part of the mitochondrial ATP synthase and thus
crucial for a correct growth of the parasite forms. A new reference related to this has
been now included (Zikova et al., PLoS Path 2009). Other antigens from where B cell
and CD8 T cell epitopes emerged were also found to have protein homologues with

high annotation scores and experimental evidences of expression. All this has been now
included in the manuscript to further support the conclusions and make our analysis
more applicable (see page 12, lines 311 – 330).

Round 2

Reviewer 2 Report

This version is improved a lot and is ready for publication.  

Reviewer 3 Report

The current manuscript is in good shape for publication.